

# New distributional records of the Samana least gecko (*Sphaerodactylus samanensis*, Cochran, 1932) with comments on its morphological variation and conservation status

Germán Chávez[1,2], Miguel A. Landestoy T[3], Gail S. Ross[4] and Joaquín A. Ugarte-Núñez[5]

[1] Instituto Peruano de Herpetología, Lima, Perú
[2] División de Herpetología, CORBIDI, Lima, Perú
[3] Escuela de Biología, Universidad Autónoma de Santo Domingo, Santo Domingo, Dominican Republic
[4] Independent Researcher, Elko, NV, USA
[5] Knight Piésold Consulting, Lima, Peru

## ABSTRACT

We report here five new localities across the distribution of the lizard *Sphaerodactylus samanensis*, extending its current geographic range to the west, in the Cordillera Central of Hispaniola. We also report phenotypic variation in the color pattern and scutellation on throat and pelvic regions of males from both eastern and western populations, which is described below. Furthermore, based on these new data, we confirm that the species is not fitting in its current IUCN category, and in consequence propose updating its conservation status.

## INTRODUCTION

Lizards of the genus *Sphaerodactylus* (107 recognized species, *Uetz, Freed & Hosek, 2020*), have diversified remarkably on Caribbean islands, and occur in Central and Northern South America and in the Pacific Island of Cocos (*Hass, 1991*; *Henderson & Powell, 2009*; *Hedges et al., 2019*; *Hedges, 2020*). This is a clade of small geckos (geckolet) containing also one of the smallest amniote vertebrates in the world with a maximum snout-vent length of 18 mm (*Hedges & Thomas, 2001*). Likewise, the largest species of this genus reaches up to a maximum of 39 mm (*Barbour, 1914*; *Schwartz & Garrido, 1985*; *Fong & Diaz, 2004*; *Griffin et al., 2018*).

The diversity of these geckos is one of the most highest among the herpetofauna in the Antilles (*Scantlebury et al., 2011*). Nonetheless, nearly 20% of the species are known only for the type locality (*Meiri et al., 2017*) and several others for a small number of localities (*Hedges, 1996*; *Powell & Inchaustegui, 2009*; *Schwartz, 1970*; *Schwartz & Henderson, 1991*). Among them, *Sphaerodactylus samanensis* is a species previously reported at a few places

Corresponding author
Joaquín A.
Ugarte-Núñez,
jugarte@knightpiesold.com,
joaqugarte@msn.com

near to the type locality, along the southern side of the Samana Bay, Dominican Republic with an elevation range from 0 to 181 m.a.s.l. (*Schwartz & Henderson, 1991*; *Thomas & Hedges, 1993*; *Landestoy, Incháustegui & Hedges, 2016*). Because its restricted distribution range and small extent of occurrence (100 km$^2$), *S. samanensis* is currently classified as a Critically Endangered species by both the IUCN Red List (2020), and the Dominican Republic's Red List of threatened species (*Ministerio del Medio Ambiente y Recursos Naturales de la República Dominicana, 2019*). According to the records, this species inhabits the northeastern edge of the island (Fig. 1) alongside Cordillera Oriental, a low mountain chain with Miocene Karst terrain (*Bowin, 1966*; *Bowin, 1975*).

The recent discovery of an individual of *Sphaerodactylus samanensis* in the surroundings of Pueblo Viejo Mine (PVM) by one of the authors (JU) encouraged us to perform new field surveys which resulted in the collection of this species at five new localities in central and eastern Dominican Republic. Our findings indicate that this species has a wider distribution than previously known, a finding relevant to its conservation status.

## Methods
### *Study area*
We conducted fieldwork at six sites (Fig. 1): (1) Caño Hondo (Los Haitises National Park), at the surroundings of the type locality and inside of its distribution range (*Landestoy, Incháustegui & Hedges, 2016*), (2) Cueva Casa Grande (western edge of the aforementioned park), and (3) Batey Piedra, all of them on the eastern edge of the Dominican Republic; and (4) Chacuey Abajo, (5) Cueva de Sanabe (inside Aniana Vargas National Park), and (6) Pueblo Viejo Mine (PVM), these last three on the Cordillera Central to the west. Eastern sites are placed on the northern slopes of the Cordillera Oriental, in the Ombrophile Rainforest (*Hager & Zanoni, 1993*), which is adjacent to the Samana Bay and goes along the Yuna River basin. The habitat here is a secondary forest, where trees, mainly *Anthoxylum* spp, *Cecropia* spp, *Chrysophyllum* spp, *Dendropanax* spp, and *Guarea* spp, reach up to 30 m in height. Also, some climber plants of the genus *Aristolochia* spp, *Cissus* spp, *Entada* spp, and *Macdafayenia* spp, and members of Orchidaceae and Poaceae family are present. The ground is densely covered by wet leaf litter and organic material, as well as scattered karst-rock clusters. Sites on the Cordillera Central were located on the easternmost border of the mountain ridge where streams flow down to the Yuna River and eventually reach the Atlantic Ocean. The landscape mainly features farms and small patches of tropical secondary rainforest with trees of the genus *Anthoxylum* spp and *Cecropia* spp reaching 35 m in height. The ground is covered partially in wet leaf litter and organic material, as well as karst-rock clustered areas. The highest altitude reached in our study is 257 m.a.s.l. in the Central Cordillera, with the lowest spot at sea level along the Samana Bay.

### *Fieldwork*
We carried out three field trips under permission number 004080 issued by the Dominican Republic's Ministry of Environment and Natural Resources (Ministerio de Medio Ambiente y Recursos Naturales - MIMARENA). Specimens were collected between August 2018 and May 2019 during diurnal surveys. We took coordinates with a personal navigator (Garmin Map 64s) and described habitat characteristics at each collection site. Every collected

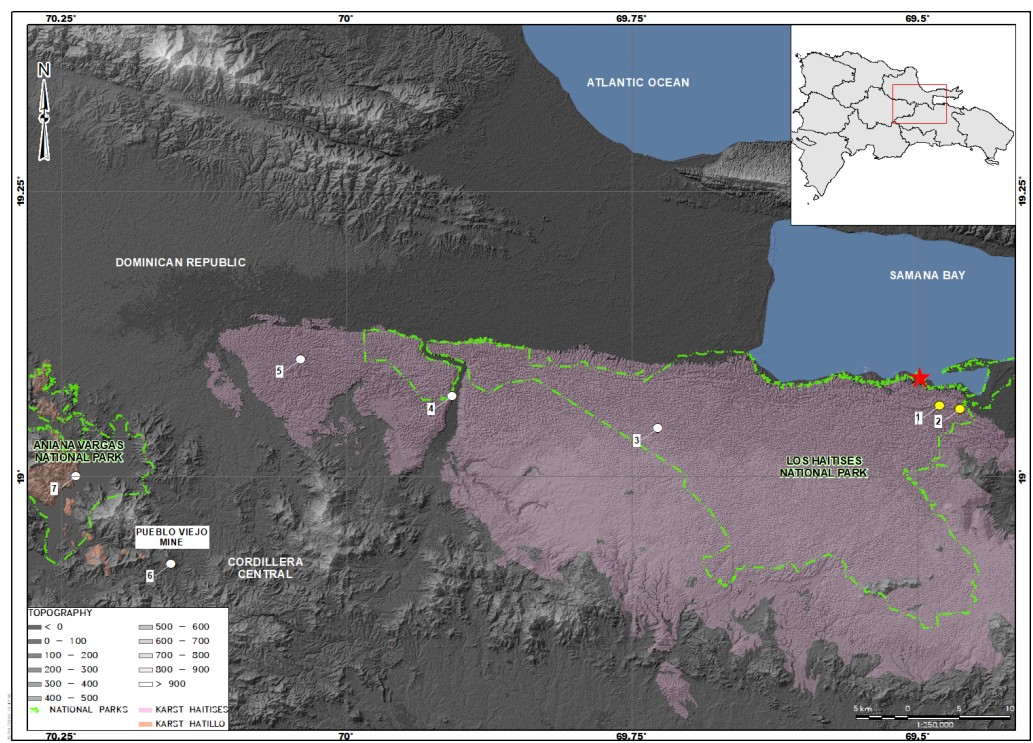

**Figure 1 Map showing the distribution of *Sphaerodactylus samanensis*.** Type locality is indicated by a red star. Localities with previous records are in yellow circles (taken from *Thomas & Hedges, 1993*; and *Landestoy, Incháustegui & Hedges, 2016*) and new collecting sites are in white circles.

specimen was photographed, measured, fixed with 95% ethanol and then stored in 70% ethanol. All the specimens were deposited in the herpetology collection of the Museo Nacional de Historia Natural Profesor Eugenio de Jesús Marcano (MNHNSD) in Santo Domingo, Dominican Republic.

*Morphological revision*

We euthanized the specimens in the field with Lidocaine 10%, fixed with 95% ethanol and then stored in 70% ethanol. We used a digital calliper to measure snout-vent length (SVL) of individuals to the nearest tenth of a millimeter. Our scale counts follow *Thomas & Schwartz (1966)* and *Thomas, Hedges & Garrido (1992)* and consists in: (1) escutcheon patch length, we considered the maximum number of scales (anterior to posterior); (2) escutcheon width, we considered the maximum number of scales transversally across the patch (including extensions onto thighs); and (3) escutcheon total scales, we considered all scales on the pelvic scutcheon. In order to support our observations, we added two more scale counts: (1) number of gular scales in contact with the first infralabial, here we considered all adjacent scales (including postmentals) to the first infralabial scale; and (2) number of scales per dorsal band, we considered the maximum number of pigmented scale rows covered by a dorsal band in a longitudinal count. Specimens were sexed by examining the sexually dimorphic color pattern and the gonads to confirm the presence of
hemipenes. We used photographs taken in the field by ML to describe the coloration in life of the specimens. Also, we followed *Köhler (2012)* to name the colors in our description. In addition, we follow taxonomy established by *Poe (2013)* which unlike (*Köhler et al., 2019*), regards *Anolis* as a valid genus for Dactyloid lizards from Hispaniola.

### Data analysis

We estimated the occurrence of this species based on our field measurements of the extension of Karst (where we observed *Sphaerodactylus samanensis*), additionally supported by the estimation of the area of Karst in contact with them, through the data previously reported by *Servicio Geológico Nacional (2010)*. Geographic data and map designing were drawn in ArcGIS version 10.3. Additionally, we follow *IUCN (2001)* defining: (1) Extent of occurrence (EOO) and (2) Area of occupancy (AOO).

## RESULTS

We observed this species mostly under Karst rocks and out of the surroundings of the type locality for the first time (specimens collected per locality are detailed in Table 1). Subsequently, we are confirming its occurrence in the Cordillera Central and adding five localities to its currently known occurrence (Fig. 1). This extends its geographic range by 82.2 km to the northwest. All individuals were observed by day, under rocks in habitat mixed between karst-rock clusters and tropical forest, with bushes and trees approaching 30 m tall, ground covered in leaf-litter and rocks covered with moss, lichens, ferns and other epiphytes. Additionally, we recorded two other geckolets: *Sphaerodactylus darlingtoni* and *S. difficilis* in sympatry with *S. samanensis*. Other sympatric lizards recorded during surveys were *Celestus sepsoides, C. stenurus, Anolis cybotes,* and *A. distichus*

All individuals of *S. samanensis* agree with the original description (*Cochran, 1932*) bearing a combination of the following characters: a moderately short snout, a large rostral scale with a median groove, a medium-sized superciliar spine, a large third supralabial exceeding the center of the eye, imbricate-keeled dorsal scales and an orange head in males. Nevertheless, we noted some phenotypic variation between *S. samanensis* individuals from the surroundings of the type locality (Caño Hondo) and nearby eastern places (Cueva Casa Grande and Batey Piedra), and the western populations (Chacuey Abajo, Cueva de Sanabe, and PVM) (See Fig. 2). The eastern individuals have 2.5–5.5 (average = 4.1, SD =0.8) gular scales in contact with first infralabial instead of 4.5–7 (average =5.1, SD =0.6) in western individuals ($p < 0.001$) (See Fig. 3), and a lower total number of pelvic scutcheon scales ranging from 25–32 scales (average =28.4, SD =2.5) instead of 30–39 scales (average =35.7, SD =2.9) in western specimens ($p < 0.001$). Eastern populations also differ in coloration by bearing dorsal bands and scapular ocelli in females and most males, which are absent in males of western samples (Fig. 2). Eastern females have 3–4 dorsal bands vs 4–5 in western females ($p < 0.001$), and wider dorsal bands covering 3–7 dorsal scales (average =5, SD =1.2) instead of the thin dorsal bands of western females covering only 3 dorsal scales (average = 3, SD = 0; $p < 0.001$). Further details on measurements, coloration and scutellation are provided in Table 2.

Chávez et al. (2021), *PeerJ*, DOI 10.7717/peerj.10404

**Table 1** **Voucher codes of *Sphaerodactylus samanensis*'s specimens collected at six localities of the Dominican Republic in this study.**

| Locality | Province | Coordinates (lat, lon) | Altitude (m) | Specimens voucher | |
|---|---|---|---|---|---|
| | | | | **Males** | **Females** |
| Caño Hondo (Los Haitises National Park) | Hato Mayor | 19.05894, −69.4633 | 44 | MNHNSD 23.3715 −16, 23.3718 | MHNHSD 23.3717, 23.3719 −20, 23.3722, 23.3893 |
| Cueva Casa Grande | Monte Plata | 19.04214, −69.72787 | 225 | MNHNSD 23.3723 | MNHNSD 23.3724 −26, 23.3894 |
| Batey Piedra | Sanchez Ramirez | 19.06997, −69.90815 | 35 | MNHNSD 23.3895 −96, 23.3899 | MNHNSD 23.3729 −31, 23.3897 −98, 23.3900 −02 |
| Chacuey Bajo | Sanchez Ramirez | 19.10689, −70.04149 | 115 | MNHNSD 23.3733 −35, 23.3905 | MNHNSD 23.3736, 23.3903 −04, 23.3906 −08 |
| Pueblo Viejo Mine | Sanchez Ramirez | 18.92348, −70.15423 | 195 | MNHNSD 23.3699, 23.3706 −07, 23.3909 | MNHNSD 23.3697 −98, 23.3701 −05, 23.3910 −14 |
| Cueva de Sanabe (Aniana Vargas National Park) | Sanchez Ramirez | 19.00004, −70.23809 | 257 | MNHNSD 23.3713 | MNHNSD 23.3712 |

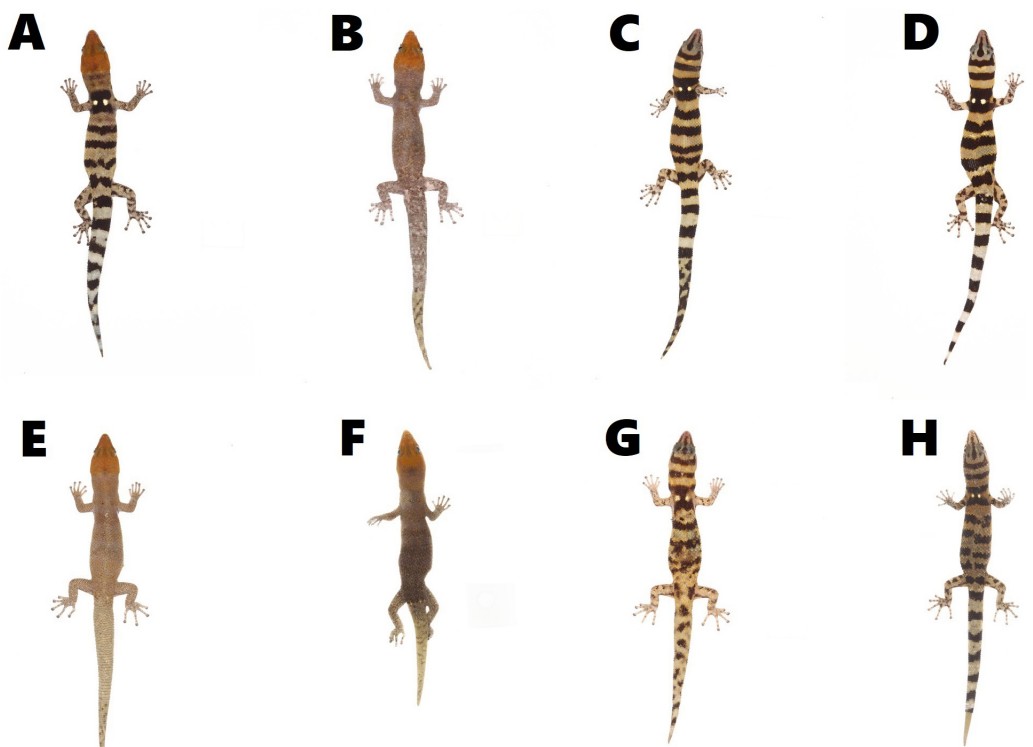

**Figure 2 Color pattern variation in *Sphaerodactylus samanensis* between Eastern and Western specimens.** Eastern males (A) MNHNSD 23.3718 (SVL = 26.3 mm), (B) MNHNSD 23.3723 (SVL = 28.6 mm), and females (C) MNHNSD 23.3717 (SVL = 27.8 mm); (D) MNHNSD 23.3719 (SVL = 27.9 mm); and Western males (E) MNHNSD 23.3733 (SVL = 27.5 mm), (D) MNHNSD 23.3713 (SVL = 24.8 mm), and females (E) MNHNSD 23.3736 (SVL = 27 mm), (F) MNHNSD 23.3712 (SVL = 26.9 mm). Photographs by Miguel A. Landestoy T.

## DISCUSSION

Our results update the distribution of *Sphaerodactylus samanensis* which now range from the region of the type locality (Boca del Infierno) in the Samana Bay (*Cochran, 1932*) and surrounding areas (*Thomas & Hedges, 1993*; *Landestoy, Incháustegui & Hedges, 2016*) to the Cordillera Central (Fig. 1), an east–west airline distance of 82.2 km. Therefore, the distribution of this gecko is now only exceeded by those of *S. copei, S. darlingtoni, S. difficilis,* and *S. elegans* (*Schwartz & Henderson, 1991*; *Hedges, 2020*), species previously recognized as widely spread on Hispaniola (*Hass, 1991*; *Schwartz & Henderson, 1991*). We also report the maximum altitude so far recorded for this species: 257 m. a. s. l. exceeding by 200 m former records reported by *Cochran (1932)* and *Landestoy, Incháustegui & Hedges (2016)*. These novel geographic data exceed those formerly known for this species confirming that it is not a short-ranged species but rather a widely distributed lineage that could be distributed even further. Additionally, an undergoing phylogenetic analysis of *Sphaerodactylus* using molecular data (B. Hedges, 2019, unpublished data) has found that population differentiation between the sampled localities is less than 1.5% confirming that our samples belong to the same wide-ranging species.
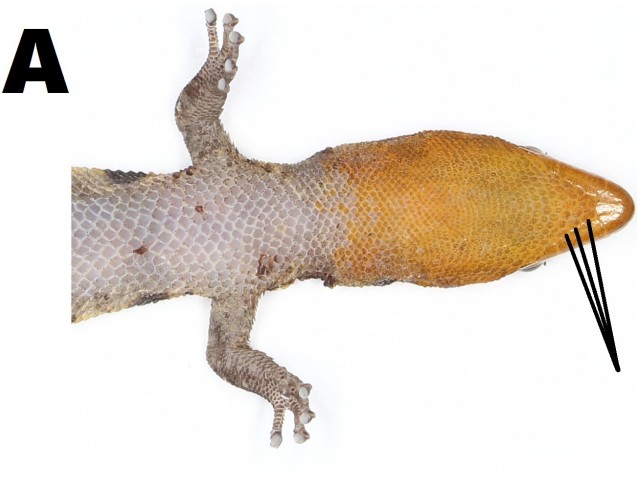

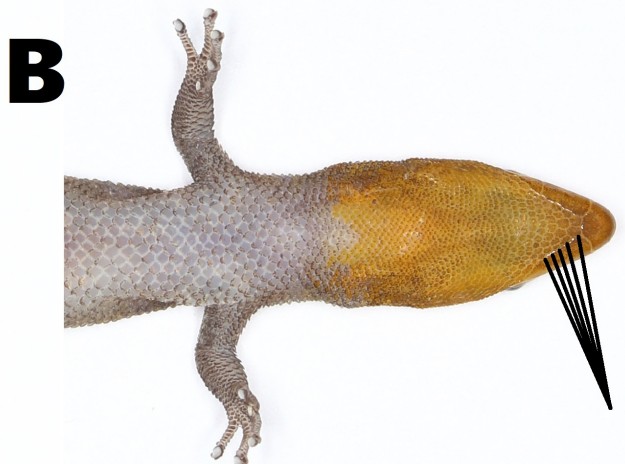

**Figure 3** **Variation in the size of gular scales (pointed with black lines) of *Sphaerodactylus samanensis.*** (A) Eastern male (MNHNSD 23.3716) from Caño Hondo, (B) western male (MNHNSD 23.3734) from Chacuey Abajo. Photographs by Miguel A. Landestoy T.

Since large geographic ranges are scarcely recorded in *Sphaerodactylus* lizards, phenotypic variation has been barely noted and subsequently poorly studied (*Schwartz, 1966*; *Dood Jr & Ortiz, 1984*). Here we provide for the first time evidence of differences between eastern (including type locality) populations ($n = 24$) and western populations ($n = 28$), mainly in color pattern and scutellation (Table 2). Measurements did not differ. In spite of scutellation mostly overlapping between eastern and western populations, gular scales are longer in eastern individuals, better noted in the proximal rows of the throat (including postmentals) which have contact with the first infralabial and are clearly smaller in western individuals (Table 2, Fig. 3). Likewise, the escutcheon plate in western males tends to contain more scales than those from eastern individuals. The differences between scutcheon width and scutcheon length are not significant ($p = 0.7$ and $p = 0.1$ respectively). This is because the difference does not depend on the width or length of the rows, but rather the number of

additional escutcheon scales surrounding the proximal edge of the escutcheon (Table 2). Concerning coloration, eastern individuals have 3–4 wide dorsal bands (each covering 3–7 dorsal scales) which are present in all females and some males (especially in males from the type locality); contrasting with western individuals which have 4–5 thin dorsal bands (each covering three dorsal scales) only present in females.

The geological history of the island of Hispaniola is influenced mainly by water incursions and plate movements occurring since the late Mesozoic and into the Cenozoic (*Mann, Draper & Lewis, 1991*; *MacPhee & Iturralde-Vinent, 1994*; *Hedges, 1996*; *Iturralde-Vinent & MacPhee, 1999*; *Ricklefs & Bermimgham, 2008*; *Daza et al., 2019*). This likely originated the vicariance phenomenon in the Proto Antilles as well as the overwater dispersion and later (approximately during Mid-Tertiary *sensu Hedges, 1996*) divergence of lineages in vertebrate fauna on this island (*Mann, Draper & Lewis, 1991*; *Hedges, Hass & Maxson, 1992*; *Hedges, 1996*; *Daza et al., 2019*). These events could cause isolation (*Hedges, 1996*; *Daza et al., 2019*) and the subsequent geographic restriction of emergent taxa to small areas, explaining why very few *Sphaerodactylus* species had been able to spread widely on Hispaniola. Those geologic events could have influenced dispersion and also the evolution of phenotypic features of *Sphaerodactylus samanensis*. Certainly, the distribution of this species seems to follow a geologic pattern overlapping two ancient karst formations (Fig. 1): Los Haitises karst to the east and El Hatillo karst to the west, both structures raised in the Late Tertiary (*Servicio Geológico Nacional, 2010*). This would agree with the phenotypic variation reported here, which follows an east–west geographic pattern. Future research should target molecular analysis and the revision of new specimens to determine patterns in the phenotypic variation in *S. samanensis.*

Because of its restricted range of distribution and threats to its habitat, both the Dominican Republic and IUCN Red-Lists currently list *Sphaerodactylus samanensis* as a Critically Endangered species (IUCN 2020). Nevertheless, our findings demonstrate that the occurrence of *Sphaerodactylus samanensis* is wider than previously reported, with an estimated EOO of 500 km$^2$. We observed that *S. samanensis* inhabits karst rocks, in contrast to sympatric congeners such as *S. darlingtoni* and *S. difficilis* which are more often recorded in leaf litter usually on soil, reducing therefore its AOO within this range. We also suggest that loss of karst formations, in particular loss of tree cover within karst areas, could threaten some populations. Nonetheless, given its widened extent of occurrence, including its presence in protected areas (Los Haitises National Park to the east and Aniana Vargas National Park to the west), as well as the number of locations and mature individuals observed during fieldwork, we propose that the species be reclassified by the IUCN. Certainly, based on new information it would appear unlikely that the species would become extinct barring catastrophic climate events, however, continued destruction of karst habitat could become a future problem for the species, therefore we propose the category Near Threatened for *S. samanensis.*

## CONCLUSIONS

Our report allows us to confirm that *Sphaerodactylus samanensis* is a widely distributed species that inhabits both eastern forests and Cordillera Central of Hispaniola Island,

Chávez et al. (2021), PeerJ, DOI 10.7717/peerj.10404

**Table 2  Color pattern, measurements (in mm) and scutellation of both eastern and western populations of *Sphaerodactylus samanensis*.**

| | Eastern Population | | Western Population | |
|---|---|---|---|---|
| | **Males ($n = 7$)** | **Females ($n = 17$)** | **Males ($n = 9$)** | **Females ($n = 19$)** |
| Coloration of the Ocular Halo | Pearl/ Bluish pearl | Copper yellow | Pearl blue/Pearl | Pearl yellow/copper yellow |
| Scapular ocelli | Present/Absent | Present | Absent | Present |
| Coloration of dorsal bands | Dark Brown/ Yellowish brown | Dark brown | — | Dark brown |
| SVL (mm) | 25.1–28.6 (26.8 ± 1.18) | 14.6–28.1 (22.3 ± 5.0) | 24.8–28.1 (26.7 ± 0.9) | 21.0–29.7 (26.1 ± 2.1) |
| Number of head stripes | 0–0 (0 ± 0.0) | 2–4 (3.8 ± 0.4) | 0–0 (0 ± 0) | 2–4 (3.8 ± 0.4) |
| Number of neck bands | 0–1 (0.5 ± 0.5) | 1–1 (1.0 ± 0) | 0–0 (0 ± 0) | 1–2 (1.2 ± 0.4) |
| Escutcheon scales (length) | 3–4 (3.2 ± 0.4) | 0–0 (0 ± 0) | 3–5 (3.7 ± 0.6) | 0–0 (0 ± 0) |
| Escutcheon scales (wide) | 10–13 (11.5 ± 0.9) | 0–0 (0 ± 0) | 10–17 (12.2 ± 2.0) | 0–0 (0 ± 0) |
| Escutcheon scales (total) | 25–32 (28.4 ± 2.5) | 0–0 (0 ± 0) | 30–39 (35.7 ± 2.9) | 0–0 (0 ± 0) |
| Number of dorsal bands | 0–4 (2.8 ± 1.9) | 3–4 (3.1 ± 0.3) | 0–0 (0 ± 0) | 3–6 (4.3 ± 0.6) |
| Number of scales per dorsal band | 0–6 (3.8 ± 1.8) | 3–7 (5.0 ± 1.2) | 0–0 (0 ± 0) | 3–3 (3.0 ± 0) |
| Number of scales in contact with 2nd infralabial | 2.5–5.5 (4.1 ± 0.8) | | 4.5–7 (5.1 ± 0.6) | |

also confirms its presence in five previously unreported localities. Likewise, we observed a phenotypic variation between eastern and western populations, however this is not consistent enough to consider them different taxon but rather this is evidence of an inter-population variation of *S. samanensis*. Moreover, our findings extend the area of occupancy of the species and lead us to suggest that its current IUCN category (CR) is not fitting with these novel data, instead we propose Near Threatened as a proper IUCN category.

## ACKNOWLEDGEMENTS

We are grateful to all whom provided field work advice, as well as reviewers for their helpful comments. We specially thank to Isabel Garcia-Cuenca by her logistic support in the field. This work was possible to the value fieldwork of our teammates Robert Ortíz, Cristian Marte, Francis Ortíz, Yimell Corona, Francis Rodríguez, Reveca Ramírez, and Pedro J. Araujo. JU thanks to Sinthya Mejia by her value feedback and Dila Valiente for designing the map.

### Funding
This work was supported by Barrick Pueblo Viejo (PO_3549_227678_0_US).

### Grant Disclosures
The following grant information was disclosed by the authors:
Barrick Pueblo Viejo (PO_3549_227678_0_US).

### Competing Interests
Joaquín A. Ugarte-Núñez is employed by Knight Piésold Consulting.

### Author Contributions
- Germán Chávez performed the experiments, analyzed the data, prepared figures and/or tables, authored or reviewed drafts of the paper, and approved the final draft.
- Miguel A. Landestoy T performed the experiments, analyzed the data, prepared figures and/or tables, and approved the final draft.
- Gail S. Ross conceived and designed the experiments, performed the experiments, analyzed the data, authored or reviewed drafts of the paper, and approved the final draft.
- Joaquín A Ugarte-Núñez conceived and designed the experiments, performed the experiments, analyzed the data, prepared figures and/or tables, authored or reviewed drafts of the paper, and approved the final draft.

### Animal Ethics
The following information was supplied relating to ethical approvals (i.e., approving body and any reference numbers):

Ministerio de Medio Ambiente y Recursos Naturales de República Dominicana approved the study (Autorización N° 004080).

## Data Availability

All the Sphaerodactylus samanensis's specimens are stored in the Museo Nacional de Historia Natural "Prof. Eugenio de Jesús Marcano" in Santo Domingo, Dominican Republic. The codes of the specimens are available in a Supplemental File.

## Supplemental Information

Supplemental information for this article can be found online at http://dx.doi.org/10.7717/peerj.10404#supplemental-information.

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
