# Peer review of "New distributional records of the Samana least gecko (Sphaerodactylus samanensis, Cochran, 1932) with comments on its morphological variation and conservation status"

_PeerJ, doi:10.7717/peerj.10404_

## Round 0.1 · original submission · Major Revisions

As you will see below, both reviewers have positive comments about your work, which is encouraging.

Could you please address the reviewer comments in your revision, especially the comment on a potential new species raised by reviewer 2 will be important to address and think about.

·

Basic reporting

This is a very simple and straightforward paper about new records from an endemic gecko from the Dominican Republic. The data reported on variation is important, but one of the photographs of the gular area can probably improved. The new records are important, especially to assess its conservation status. I guess since the new records are from protected areas, this is great news for the species. The figures are good, but see my comment on the gular scales. Can you do a close up of the postmental scales? These seem to be more numerous in the eastern populations.

Experimental design

This section is adequate for the type of research. Perhaps doing some box plots might help showing better the differences among populations.

Validity of the findings

Sphaerodactylus geckos are species very hard to find a collect. This paper is important for the conservation status of a poorly know species. I think the geological discussion is relevant to understand orogenic process in the area, but the age of this group is newer, so I don't think it has a lot of relevance for its distributional range. In this regard, maybe some niche models might me more relevant, or to determine what environmental factors are important, temperature, precipitation, etc.

Additional comments

This is a very simple paper, I like the general paper, but I would like to see more ecological data related to the habitat before this paper is published. The botanical assessment of the area is very vague, and basically would apply to many areas in the Caribbean. I recommend including additional microhabitat information about this species, which would be probably very useful (e.g. species of trees on the area, is the leaf litter moist or dry, is the leaf litter dense or sparce, etc) . You mention these geckos were found under rocks, would you classify the as rupiculous, or a combination of leaf litter dwellers using occasionally karst outcrops?

Reviewer 2 ·

Basic reporting

no comment

Experimental design

no comment

Validity of the findings

see below

Additional comments

Reviewer Comments for "New distributional records of the Samana least gecko (Sphaerodactylus samanensis, Cochran, 1932) with comments on its morphological variation and conservation status"

The main focus of the manuscript is the discovery of previously undocumented S. samanensis localities, which expands the known distribution of the species and presumably warrants a re-evaluation of its conservation status as Critically Endangered. While this seems like a straightforward conclusion there are several things that make me question this line of thinking. First, a recent molecular phylogeny, that included many Hispaniolan species of Sphaerodactylus (Daza et al. 2019. The sprightly little sphaerodactyl: Systematics and biogeography of the Puerto Rican dwarf geckos Sphaerodactylus (Sphaerodactylidae, Gekkota). Zootaxa 4712:151–201) found S. ladae, S. difficilis, & S. darlingtoni were actually species complexes composed of more than one valid taxon. Thus, multiple widespread species of Hispaniolan Sphaerodactylus are in need of taxonomic revision and that could also include the now widespread S. samanensis. Analysis of molecular data would be helpful in this case to determine whether the eastern and western S. samanensis populations are reproductively isolated. Second, the authors detail significant phenotypic variation between the eastern and western S. samanensis populations. These morphological differences, when considered together, are sufficient to diagnose each population and perhaps even call them separate species. Third, the authors describe a biogeographic scenario (lines 174-190) that that could have resulted in allopatric speciation between eastern and western S. samanensis populations. Thus, while S. samanensis clearly has a larger distribution than previously known, it may not be as big as stated in this manuscript and the authors in all likelihood have discovered an undescribed species, closely related to S. samanensis sensu stricto. Thus, reconsidering the conservation status of S. samanensis, while clearly warranted, must be contingent upon a thorough taxonomic evaluation first.

Below are some additional comments that I hope the authors find useful.

Lines 52-54 - Should also mention that 21 Sphaerodactylus species are known only from the type locality (Meiri et al. 2017. Extinct, obscure or imaginary: The lizard species with the smallest ranges. Diversity & Distributions 24:262-273).

Lines 109-110 - Perhaps cite (Poe. 2012. 1986 Redux: New genera of anoles (Squamata: Dactyloidae) are unwarranted. Zootaxa 3626:295–299) as justification for using Anolis.

Line 158 - Should be Dodd Jr. & Ortiz (see also line 225).

Line 168 - "(intruders)" - not sure what this refers to.

---

## Round 0.2 · Minor Revisions

You will see that the reviewer only had a few additional comments that they would like you to address. If you could respond to these comments, I'd greatly appreciate it.

·

Basic reporting

The manuscript has improved considerably compared with the previous version, However there are a few things that still are unclear and need to be addressed. See comments to authors section.

Experimental design

NA

Validity of the findings

The content of this manuscript is important and the data is valid.

Additional comments

Thanks for sending a revised version.

I am sorry to insist but dominant needs to be specified, is this in terms of biomass population density, or biodiversity? Certainly, Anoles are probably more diverse than sphaeros.

In the part where you write the species of trees, do not writte the abbreviation of species italics, that is incorrect.

I understand you want to wait to publish your phylogenetic analysis on a separate analysis, but the comment from the other reviewer is still valid. By the way, I could not find the tree mentioned in the rebuttal or a reference to it as supplementary material on the text. What I suggest you can do is to write a brief statement like the one below and make reference to the genetic distance of these populations:

An undergoing phylogenetic analysis of Sphaerodactylus from ______________ using molecular data has gound that population differentiation between the sampled localities is less than XX%.

---

## Round 0.3 · accepted · Accept

Thank you for making the changes requested by the reviewer.